# The Modulatory Effect of Selol (Se IV) on Pro-Inflammatory Pathways in RAW 264.7 Macrophages

**DOI:** 10.3390/ijms26020559

**Published:** 2025-01-10

**Authors:** Gwan Yong Lim, Emilia Grosicka-Maciąg, Maria Szumiło, Daniel Graska, Iwonna Rahden-Staroń, Dagmara Kurpios-Piec

**Affiliations:** 1Chair and Department of Biochemistry, Medical University of Warsaw, Banacha 1, 02-097 Warszawa, Poland; s083854@student.wum.edu.pl (G.Y.L.); mszumilo50@gmail.com (M.S.); danielgraska@outlook.com (D.G.); iwonnarahden@gmail.com (I.R.-S.); dagmara.kurpios-piec@wum.edu.pl (D.K.-P.); 2Department of Biochemistry and Laboratory Diagnostics, Faculty of Medicine, University of Cardinal Stefan Wyszyński, Wóycickiego 1/3, 01-938 Warszawa, Poland

**Keywords:** Selol, oxidative stress, inflammation

## Abstract

Selol is a semi-synthetic mixture of selenized triglycerides. The results of biological studies revealed that Selol exhibits several anticancer effects. However, studies on its potential anti-inflammatory activity are scarce, and underlying signaling pathways are unknown. The aim of our study was to investigate the ability of Selol to exert anti-inflammatory effects in a RAW 264.7 cell line model of LPS (lipopolysaccharide)-induced inflammation. Cells were treated either with Selol 5% (4 or 8 µg Se/mL) or LPS (1 µg/mL) alone or with Selol given concomitantly with LPS. The parameters studied were reactive oxygen species (ROS) production, glutathione and thioredoxin (Txn) levels, and nuclear factor kappa B (NF-κB) activation, as well as nitric oxide/prostaglandin E_2_ (NO/PGE_2_) production. The presented research also included the effect of Selol and/or LPS on glucose (Glc) catabolism; for this purpose, the levels of key enzymes of the glycolysis pathway were determined. The results showed that Selol exhibited pro-oxidative properties. It induced ROS generation with a significant increase in the level of Txn; however, it did not affect the reduced glutathione/oxidized glutathione (GSH/GSSG) ratio. Selol moderately activated NF-κB but failed to affect NO/PGE_2_ production. The effect of Selol on glucose catabolism was not significant. However, the simultaneous administration of Selol with LPS exerted a statistically significant anti-inflammatory effect via a decrease in the production of pro-inflammatory mediators and NF-κB activation. Our study also showed that as a result of LPS action in cells, the anaerobic glycolysis activity was increased, and incubation with Selol caused a partial reprogramming of Glc metabolism towards aerobic metabolism. This may indicate different pharmacological and molecular effects of Selol action in physiological and pathological conditions.

## 1. Introduction

Inflammation is a tightly regulated process that normally serves to recruit the immune cells to sites of infection or injury and to restore tissue function and structure. In some cases, excessive and inappropriate inflammatory activity may lead to chronic inflammation. Chronic inflammation is a primary contributing factor in many chronic diseases, such as cardiovascular diseases, atherosclerosis, type 2 diabetes, rheumatoid arthritis, and cancer [1,2,3,4,5].

Macrophages play a significant role in the initiation, maintenance, and resolution of inflammation [6,7,8,9]. After activation of macrophages by interferon-γ (IFN-γ) or LPS, several signaling cascades are initiated including interferon regulatory factor 3 (IRF3), NF-κB, and mitogen-activated protein kinase (MAPK) pathways [10,11]. In consequence, inflammatory mediators such as NO, PGE_2_, tumor necrosis factor-α (TNF-α), and interleukin-1β (IL-1β) are produced in high amounts and contribute to the inflammatory response [12,13,14]. Another aspect of the inflammatory process is the metabolic reprogramming of macrophages during the activation/initiation and resolution of inflammation [15,16,17]. Both initiation and resolution are high-energy-demanding reactions, and they require the temporary adaptation of cell metabolism to support an adequate amount of ATP. After stimulation with LPS, macrophages exhibit a glycolytic metabolic profile with concomitant decreases in oxidative phosphorylation; moreover, activated macrophages decrease the expression of some enzymes of the Krebs cycle [18,19,20].

The beneficial and multidirectional effects of selenium (Se) on human health encouraged clinicians to investigate the potential use of this element for the treatment of multiple diseases [21]. Selenium plays an important role in the defense against pro-oxidants, the proper functioning of the immune system, and as the component of several metabolic pathways [22]. It is not only involved in initiating or enhancing immunity, but it is essential for immunoregulation. Se deficiency may have negative impact on the activation, differentiation, and proliferation of immune cells [20,23]. The recent studies of Korwar et al. [20] showed that supplementation with Se causes an extensive reprogramming of energetic metabolism in BMDMs (bone marrow derived macrophages) upon LPS stimulation. The tested cells exhibit intensified activity in the Krebs cycle and pentose phosphate pathway, as well as oxidative phosphorylation. This allows them to undergo a phenotypic transition toward alternatively activated macrophages comparable with the resolution of inflammation. It might be hypothesized that this metabolic reprogramming after Se supplementation regulates inflammation and its timely resolution.

Among many Se compounds, the highest activity as anti-oxidants and anticancer agents was demonstrated by those containing tetravalent Se (IV). Selol, one of them (Figure 1), is a semi-synthetic mixture of selenized triglycerides, which are produced by the reaction of selenite with sunflower oil. It was characterized by high-performance liquid chromatography–inductively coupled plasma mass spectrometry (HPLC-ICP MS) and high-performance liquid chromatography–electrospray ionization tandem mass spectrometry (HPLC ESI MS/MS) [24]. Se in the form of Selol demonstrates lower toxicity than inorganic sodium selenite, and it does not reveal any cumulative toxicity or mutagenicity [25,26]. Previous studies have shown anticancer properties of Selol connected, among others, with the disruption of the redox regulation in androgen-dependent hPCa-bearing mice and significantly lower activity of certain anti-oxidant enzymes in malignant cells [27,28,29,30,31,32,33,34,35]. Other studies have shown that supplementation with 5% Selol affects the activities of selenoenzymes and the anti-oxidant status in plasma, erythrocytes, and organs of healthy mice [26,36]. The recent results obtained by Sochacka et al. 2024 revealed a promising strategy for the anticancer activity of Selol in an in vivo model [37]. Long-term supplementation with Selol improved the anti-oxidant mechanism in healthy mice but induced oxidative stress in tumor cells with concomitant activation of their anti-oxidant mechanisms. They also proposed a unexpected hypothesis about the anticancer properties of Selol. It seems to induce the death of cancer cells in small tumors; however, in advanced tumors, it may stimulate tumor growth when is used as a monotherapy. Also, results presented by Pinherio et al. demonstrated the anticancer properties of Selol [38,39]. Selol, in the form of nanocapsules, inhibited breast tumor growth in an animal model.

Because studies on Selol’s anti-inflammatory activity are scarce, and underlying signaling pathways are still unknown, we aimed to investigate the ability of Selol to exert an anti-inflammatory response. We used the LPS-induced mouse macrophage cell line (RAW 264.7) as a model of inflammation. Simultaneously, we examined how Selol affects energy metabolism and glutathione concentration in control RAW 264.7 and under conditions of inflammation.

We showed that Selol alone has a pro-oxidant effect through the induction of ROS generation and moderate activation NF-κB, but it failed to affect NO/PGE_2_ production. However, the simultaneous administration of Selol with LPS exerted a statistically significant but weak anti-inflammatory effect via a decrease in the production of pro-inflammatory mediators and NF-κB activation. This may indicate different pharmacological and molecular effects of Selol action in physiological and pathological conditions.

## 2. Results and Discussion

### 2.1. Effect of Selol on RAW 264.7 Viability

Based on our previous results [40], we selected the following concentrations—2, 4, and 8 μg Se/mL—to assess the effect of Selol on the viability of RAW 264.7 cells. Our results confirmed the low cytotoxicity of Selol. It reduced cell viability by an average of 13.4% ± 1.91. After stimulation with LPS, cell viability was decreased by 6.7% (*p* < 0.05). The concomitant treatment of cells with LPS and Selol revealed a slight increase in cell viability (Figure 2). Results of our studies confirmed the thesis that Se, in the form of Selol, exhibits low cytotoxic action in cells [40,41,42].

### 2.2. The Effect of Selol on NF-κB Activation

In our study, we examined the potential of Selol to attenuate inflammation induced by LPS in RAW 264.7 cells. To investigate the influence of Se on the inflammatory mechanism, we analyzed the critical protein NF-κB by confocal microscopy. After incubation with 4 mg Se/mL, NF-κB/p65 moved to the nucleus (Figure 3B), whereas treatment with 8 mg Se/mL led to the accumulation of NF-κB/p65 around the nuclear membrane (Figure 3C). RAW 264.7 cells exposed to LPS revealed a dense accumulation of NF-κB p65 in the nucleus (Figure 3D). The concomitant treatment of cells with LPS and Selol at both concentrations (4 or 8 µg Se/mL) resulted in a dense accumulation of NF-κB p65 in the nucleus (Figure 3E,F).

The obtained results firstly demonstrated the activation of NF-κB in response to Se in macrophages; secondly, what is more interesting is that the observed effect of Se in the inflammatory model was ambiguous. Our findings confirmed the thesis that the direction of selenium action in cells is not clear and depends on the form of Se and type of the research model. The results of the transcriptomic and proteomic datasets presented by Meplan et al. [43] revealed that the expression of 69 genes, including selenoproteins W1 and K, which are involved in cytoskeleton remodeling and NF-κB signaling, correlated significantly with Se status. The study conducted in rectal biopsies from 22 healthy adults showed reduced inflammatory and immune responses and cytoskeleton remodeling in group with the suboptimal Se status. An interesting hypothesis is presented in [44]. Based on the fact that increased Se status in cells enhances the expression of selenoproteins, including thioredoxin reductase 1 (TXNRD1), and promotes maximal DNA-binding activity of NF-κB, they propose the hypothesis that Se might activate NF-κB and downstream inflammatory pathways through improved activity of anti-oxidant selenoproteins [44]. Selenoproteins are described as redox gatekeepers because they catalyze reactions involving the reduction of disulfides, methionine sulfoxide, and peroxides [45,46,47]. Moreover, the activity of selenoproteins may improve the inflammatory response in cells by reducing oxidative stress and by the redox regulation of inflammatory signaling pathways, which lead to cytokine/chemokine production [21].

### 2.3. The Effect of Selol on ROS Production

This dual effect of Selol on NF-κB activation can be the result of its pro- and anti-oxidant properties. Flis et al. [27] revealed that Selol has an initial strong pro-oxidative and antineoplastic effect, while in the second phase of action, it exhibits anti-oxidant and repair properties. After stimulation of RAW 264.7 cells with Selol (4 or 8 μg Se/mL), green fluorescence was detected due to the increased ROS production in cytosol (Figure 4C,D). Similar results were observed in cells treated with H_2_O_2_, which was used as positive control (Figure 4A). Furthermore, this experiment showed that Selol did not exhibit cytotoxic effects. We did not detect red fluorescence characteristic of dead cells in macrophages exposed to both concentrations of Selol (Figure 4C,D).

The spectrophotometric analysis revealed that macrophages pretreated with dihydrorhodamine 123 (DHR123) or 2′,7′-dichlorodihydrofluorescein diacetate (H_2_DCFDA) exhibited significant increases in fluorescence intensity after 1 h incubation with 8 μg Se/mL (*p* < 0.05 and *p* < 0.02, respectively), compared to control cells (Figure 5A and Figure 5B, respectively). Fluorescence intensity in RAW 264.7 cells loaded with ethidium raised significantly after 3 h incubation with 8 μg Se/mL (*p* < 0.05) (Figure 5C). The observed increases in fluorescence intensity are associated with the oxidation of fluorescent dyes by ROS generated by Selol.

In the present study, Selol revealed pro-oxidative properties parallel to increased ROS production. It is well known that NF-κB is a highly redox-sensitive transcription factor; therefore, it is considered as a significant link between oxidative stress and inflammation. Both ROS and RNS (reactive nitrogen species) play important roles in its regulation. These reactive molecules may be responsible for the activation or inhibition of NF-κB, depending on many factors: the cell type, the oxidant studied, and the concentration of ROS or RNS, as well as the concomitant stimulation of cells with NF-κB activators such as cytokines or LPS together with oxidants. Moreover, the redox regulation of NF-κB may dynamically change at different stages of the inflammatory response. ROS may activate the NF-κB pathway in the early phase of inflammation, while at later stage of it, these reactive molecules may inhibit NF-κB signaling, promoting the induction of tissue repair [48,49]. Razaghi et al. [44] proposed that Se, either as a pro-oxidant or anti-oxidant, can theoretically induce a pro-inflammatory response in the cell through NF-κB activation. However, there is still a lack of knowledge about the detailed mechanisms which match the Se with NF-κB signaling pathways.

### 2.4. The Effect of Selol on Reduced Glutathione (GSH) and Oxidized Glutathione (GSSG) Concentration and Thioredoxin (Txn) Level

The results of in vitro studies revealed that Se might affect immune function by modulating ROS and/or glutathione levels [50,51]. Both Txn/TxnR (thioredoxin/thioredoxin reductase) and GSH/GSSG ratios are crucial parts of the cellular anti-oxidant system, and their operations are based on the oxidation and reduction reactions of dithiol–disulfide [52]. Our results showed that Se did not significantly affect the GSH/GSSG ratio; both reduced and oxidized glutathione were similar to that in the control cells; simultaneously, we observed significant increases in thioredoxin levels (Thioredoxin OS = Mus musculus OX = 10,090 GN = Txn PE = 1 SV = 3) in cells supplemented with Se (Table 1). Selol used at 4 and 8 mg/mL caused 1.8- and 2.15-fold increases in Txn levels, respectively.

Interesting results were obtained for cells exposed to LPS. We noticed a significant drop in the GSH/GSSG ratio (by 88.52%, compared to the control) but with concomitant increases in Txn by 2.36 folds. In macrophages exposed to LPS and Selol (4 and 8 mg/mL), the GSH/GSSG ratio was partially restored (Table 1), whereas the concentration of Txn in cells exposed to 4 mg/mL of Se together with LPS was slightly lower than in the control, but in cells exposed to 8 mg/mL of Se and LPS, the level of Txn was 1.54 times higher than in macrophages exposed to LPS alone.

It should be stressed that in addition to being a crucial component of the thioredoxin–thioredoxin reductase system, Txn possesses a strong reduction potential for enzymes and plays a role in the reversible S-nitrosylation of Cys residues in target proteins, and thereby contributes to the response to intracellular nitric oxide. Moreover, Txn regulates the activity of some crucial transcription factors such as NF-κB or AP-1. Yang et al. [53] demonstrated that Txn might be involved in regulating the expression of selenoproteins, glutathione peroxidase 1–4 (GPx1–4) and TxnR, in a chicken model. Under oxidative stress, Txn together with NF-κB moved into the nucleus. In the nucleus, Txn is responsible for the structural modification of NF-κB via the reduction of Cys residues. This step is necessary for NF-κB activation and its binding with DNA [54]. Up to now, the contribution of Txn in inflammatory responses has not been clear. Both activating and inhibitory effects have been described [55,56,57]. The influence of Se on cellular ROS scavengers’ system and NF-κB is also controversial, and it depends mainly on the Se treatment concentration. Se deficiency may cause oxidative stress by lowering the activity of leading selenoproteins and Txn. On the other hand, Se supplementation exhibits chemopreventive effects mediated by the increased expression of selenoproteins. Observed changes both in the Thx concentration and in the GSH/GSSG ratio in RAW 264.7 cells exposed to Se or LPS and Se correlate with the noticed changes in NF-κB activation. Our results confirmed the hypothesis that activity of Txn and NF-κB are closely related. Moreover, under inflammatory conditions, the influence of Se on Txn activity was dose-dependent and dual-faced. Se at 8 mg/mL significantly increased Thx activity, whereas Se at 4 mg/mL decreased the level of Txn. In recent years, it has been increasingly indicated that compounds that have an inhibitory effect on the Txn system may have therapeutic significance in cancer therapy, but up to now, the involvement of the Txn system in the induction and resolution of inflammation has not been clear.

### 2.5. The Effect of Selol on Cellular Energy Metabolism

ATP and glucose are critical molecules in the inflammatory response. Freemerman et al. [58] reported that GLUT1 is the primary rate-limiting glucose transporter and its overexpression in RAW 264.7 resulted in elevations in both glucose uptake and catabolism, with a concomitant reduction in oxygen consumption rate. These processes may independently drive a hyperinflammatory state, exemplified by the raised secretion of inflammatory mediators [58]. Other studies indicated that after induction with LPS, macrophages decrease oxidative phosphorylation and display a glycolytic metabolic phenotype like the Warburg effect in cancer cells [20]. Our results confirmed the thesis that LPS significantly affects glucose metabolism in RAW cells (Figure 6). We observed an increased expression of phoshoglycerate kinase (1.7-fold) and puryvate kinase (2.16-fold) compared to the control cells. Both enzymes are responsible for direct ATP synthesis in substrate-level phosphorylation in the glycolysis pathway. Moreover, RAW 264.7 cells presented a high expression of lactate dehydrogenase (3.3-fold compared to the control) after LPS exposure; this may indicate a switch towards anaerobic metabolism. Freemerman et al. [58] suggested that glucose metabolism is central to the function of classically activated macrophages and might be the potent target for altering inflammatory responses. In their work, they also showed that anti-oxidants reversed GLUT1-mediated pro-inflammatory elevations. In our studies, we analyzed the effect of Selol (4 or 8 μg Se/mL) on glucose metabolism in murine macrophages during inflammation. We showed that Se might have partially blunted the effects of LPS. The expression of phoshpoglycerate kinase was lower after exposure to Selol and LPS than in cells treated only with LPS, especially after exposition to Selol at 8 μg Se/mL and LPS. However, the expression of pyruvate kinase decreased only in cells exposed to Selol at 8 μg Se/mL and LPS (Figure 6). We also noticed that Selol, in both tested concentrations, reversed the effects of LPS on lactate dehydrogenase, and the expression of this enzyme was on the same level as in the control cells. This might indicate that Se leads to the reprogramming of cellular metabolism toward aerobic. Our results are in agreement with work presented by Korwar et al. [20]. They proposed that the Se-dependent modulation of main catabolic pathways toward oxidative phosphorylation might be a key factor which regulates inflammation and its timely resolution.

### 2.6. Effect of Selol on NO/PGE2 Produciton

LPS induces an inflammatory response in various cells, including macrophages, via interaction with its specific receptor, Toll-like receptor 4 (TLR4). In consequence, signal transduction pathways are activated, leading to increased cyclooxygenase-2 (COX-2) and induciblenitric oxide synthase (iNOS) expression [59]. Since Selol slightly decreased NF-κB activation in LPS-stimulated RAW 264.7 cells, we evaluated the production of PGE_2_ and NO (Figure 7 and Figure 8, respectively). We did not observe changes in either NO and PGE2 concentrations in cells exposed only to Selol. After stimulation with LPS, a 56-fold increase in NO production (99.05 ± 4.34 μM) was observed, compared to the control cells. The concomitant stimulation of RAW 264.7 cells with LPS and Selol (4 or 8 μg Se/mL) resulted in a significant decrease in NO production by 21% (*p* < 0.001) and 29% (*p* < 0.001), respectively, compared to cells exposed to LPS alone. PGE_2_ concentration in cells stimulated with LPS was 300-fold higher (6274.62 ± 243.71 pg/mL) compared to the control cells. Cells stimulated with Selol (4 or 8 μg Se/mL) together with LPS revealed a statistically significant decrease in PGE_2_ concentration by 14% (*p* < 0.001) and 21% (*p* < 0.001), respectively, compared to LPS-treated cells.

Our results are in agreement with other studies which present the beneficial effects of Se in inflammatory conditions. COX-2 is one of the key enzymes regulated by selenium [60]. The study of Zamamiri-Davis et al. [61] showed that Se supplementation affects NF-κB activation and downregulates COX-2 expression without alteration in COX-1 expression in RAW 264.7 cells stimulated with LPS. It has also been reported that the anti-inflammatory properties of Se are driven by increases in the macrophage production of arachidonic acid (AA)-derived anti-inflammatory 15-deoxy-Δ^12,14^-prostaglandin J_2_ (15d-PGJ_2_) by COX-1 and decreases in the pro-inflammatory PGE_2_ production [9,62]. Se status also affects the regulation of iNOS expression, most likely through influencing the signaling pathways involved in iNOS induction. Prabhu et al. [63] showed a significant increase in the expression of iNOS mRNA and protein in RAW 264.7 cells deficient in Se, compared to cells supplemented with Se. Moreover, the correlation of iNOS expression and NF-κB activation was observed. The recent results of Ge et al. [64] confirmed the beneficial effect of Selenium (Se), which alleviates the negative effects of cadmium (Cd) toxicity. Cd treatment enhanced iNOS activities and the expression of NOS isoforms via the NF-κB/IκB pathway to promote inflammation responses in the heart. Ge et al. demonstrated that Se inhibited the NO enzymes activities and NOS isoform alterations on Cd-induced cardiovascular inflammation via the NF-κB/IκB pathway [64]. Altogether, in the present in vitro setting, Se in the form of Selol exhibits immunomodulatory properties in immune cells both in control and LPS-induced cells. Under inflammatory conditions, it alleviated inflammatory responses in cells. Selol shows similar immunogenic properties to the lunesin peptide. It was shown to have an anti-inflammatory effect in RAW cells and intestinal mucosa both in the presence/absence of LPS. It induced the synthesis of many pro-inflammatory cytokines (TNF α, IL-1β) while reducing the expression of iNOS and p65 NF-κB. On the other hand, it alleviated LPS-induced responses in both models [65].

## 3. Materials and Methods

### 3.1. Materials

Selol was obtained through the chemical modification of sunflower oil (Polish patent no. 176,530 granted in 1999) in the Department of Bioanalysis and Drug Analysis at the Medical University of Warsaw. The mixture of 11 distinct selenium triglycerol derivatives containing 0.5–5% (*w*/*w*) of Se (+4) were identified by 1H and 13C NMR studies. The indication 0.5–5% (*w*/*w*) Selol refers to the actual content of selenium (+4) as 0.5% (*w*/*w*) to 5% (*w*/*w*), respectively. Based on the published dates, we used a micellar solution of Selol (based on lecithin, water, and Selol), which was prepared ex tempore with the declared concentration of 5% (*w*/*v*) constituents [24,25,26,66,67,68].

### 3.2. Cell Culture

RAW 264.7 cells, a murine macrophage cell line (ATCC, No TIB-71™), were obtained from the American Type Culture Collection (Teddington, UK) and cultured in DMEM medium supplemented with 10% (*v*/*v*) heat-inactivated fetal bovine serum, glucose (4.5 g/L), ultraglutamine (200 µM), HEPES buffer (20 mM), 100 U/mL penicillin, and 100 µg/mL streptomycin at 37 °C in a humidified air/CO_2_ (5%) atmosphere. Standard cultures were tested in culture medium or culture medium supplemented either with 1 μg/mL *Escherichia coli* LPS (serotype O127:B8, from Sigma-Aldrich, Saint Louis, MO, USA) or Selol (0–8 μg Se/mL) or the combination of both LPS (1 μg/mL) and Selol (0–8 μg Se/mL).

### 3.3. Cell Viability

The growth inhibition effect of Selol on cells was determined by measuring the MTT (3-[4,5-Dimethylthiazol-2-yl]-2,5-diphenyltetrazolium bromide) dye absorbance in living cells as previously described [69]. Cells (2 × 10^4^) were dispensed into 96-well plates either in culture medium, culture medium supplemented with Selol (2–8 µg Se/mL), LPS (1 µg/mL), or Selol (2–8 µg Se/mL) plus LPS (1 µg/mL) for 18 h at 37 °C. The optical density was measured with a UVM 340 (ASYS Hitech GmbH, Eugendorf, Austria) microplate reader at 570 nm.

### 3.4. ROS Detection by Confocal Microscopy

RAW 264.7 cells were grown to 70% confluence on sterile 4-well slides. Thereafter, cells were incubated in complete medium with Selol (4 or 8 µg Se/mL) for 1 h. Then, dihydrorhodamine 123 (1 μM) was added to macrophages for 30 min. Ten minutes before the end of incubation, propidine iodide (30 μM) was added to the cells. After washing the cells three times with PBS, coverslips were mounted with fluorescent mounting medium and analyzed with a Leica TCS SP5 (Leica, Wetzlar, Germany) at excitation and emission wavelengths of 500 nm/536 nm for rhodamine and 536 nm/617 nm for propidium iodide, respectively. A sample with 1.5 mM H_2_O_2_ was used as a positive control, and a sample without any reagent was used as a negative control. This concentration was selected based on the published dates, and it allowed us to obtain a clear increase in fluorescence [70,71]. Four to six slides were examined with similar results, and representative experiments are shown in the corresponding photomicrographs.

### 3.5. ROS Detection: DHR 123, DCFH-DA, and HE Assay

ROS generation was evaluated by the spectrofluorometric method using the DHR 123, DCFH-DA, or hydroethidine (HE), a sodium borohydride-reduced derivative of ethidium bromide as previously described [40]. RAW 264.7 cells (5 × 10^4^) were incubated with DHR123 (1 μM), DCFH-DA (5 μM), or HE (5 µM) on 96-well plates for 30 min at 37 °C in the dark to allow dye loading into the cells. Thereafter, the cells were rinsed with PBS and treated for 1 and 3 h with phenol red-free DMEM, which contained Selol (4 or 8 μg Se/mL). The fluorescence intensity (FI) of the cells was immediately measured using a microplate spectrofluorometer (BioTek Synergy™4, BioTek Instruments, Winooski, VT, USA), and values from at least three experiments were analyzed.

### 3.6. Colorimetric Determination of Total (GSHt) and Oxidized (GSSG) Glutathione and GSH/GSSG Ratio in RAW 264.7 Cells After Selol Treatment

RAW 264.7 cells were seeded (3 × 10^5^/well) into 6-well plates. After the cells reached 80% confluency, Selol and/or LPS were added for 18 h. Then, the cells were scraped and centrifuged at 1000× *g* for 10 min. The pellet washed with PBS was used for the determination of GSSG and GSH levels using a Total Glutathione (T-GSH)/Oxidized Glutathione (GSSG) Colorimetric Assay Kit (Elabscience Biotechnology, Wuhan, China). The results of the total (reduced and oxidized—GSHt) and oxidized glutathione (GSSG) were normalized by the cell number (1 × 10^6^) and expressed as nmol/10^6^ cells.

### 3.7. Analysis of NF-κB Induction with Laser Scanning Confocal Microscope (LSCM)

NF-κB activation in RAW 264.7 cells was examined in cells stimulated either with Selol (4 or 8 µg Se/mL), LPS (1 μg/mL) alone, or LPS plus Selol (4 or 8 µg Se/mL) for 1 h. Analysis was performed with a laser scanning confocal microscope (LSCM) as previously described [40]. On average, five to six slides were examined by LSCM with similar results, and the representative experiments are shown in the corresponding photographs.

### 3.8. LC-MS Proteome Analysis

Enzymes involved in cellular energy metabolism and thioredoxin were analyzed in cell lysates obtained after treating cells with LPS (1 μg/mL) or/and Selol (4 or 8 µg Se/mL) for 18 h. Cells were washed with phosphate-buffered saline (PBS), harvested, and centrifuged at 1000× *g* for 10 min. Lysis buffer (with protease inhibitor and 1% RIPA Lysis and Extraction Buffer (ThermoFisher Scientific, Norristown, PA, USA) and cold PBS were added, followed by sonication three times in an ice bath. Then, the cell lysates were centrifuged at 14,000× *g* at 4 °C for 15 min, and the supernatants were stored at −70 °C before use. Protein concentration was measured by the Bradford method [72]. Normalized protein concentrations (5 μg) from cell lysate were precipitated by ice-cold acetonitrile (ACN, Merck, Darmstad, Germany, in a 1:4 ratio). The samples were centrifuged (−9 °C, 30 min, 18,000× *g*), the supernatant was discarded, and excess ACN was evaporated using a vacuum centrifuge (5 min, room temperature). The protein pellet was dissolved in 40 mM ammonium bicarbonate. Reduction and alkylation processes used 500 mM dithiothreitol (DTT, final concentration 20 mM) and 1 M iodoacetamide (IAA, final concentration 40 mM). After 16 h of incubation at 37 °C with Trypsin Gold (Promega, Madison, WI, USA), the digested protein samples were diluted with 0.1% formic acid (ThermoFisher) and centrifuged (+2 °C, 30 min, 18,000× *g*). LC–MS analysis was carried out using nanoUHPLC (nanoElute, Bruker, Billerica, MA, USA) coupled with CaptiveSpray (Bruker) and an ESI-Q-TOF mass spectrometer (Compact, Bruker). A two-column separation method was used: a pre-column (300 μm × 5 mm, C18 PepMap 100, 5 μm, 100 Å, ThermoFisher Scientific, Norristown, PA, USA) and an Aurora separation column with CSI fitting (75 μm × 250 mm, C18 1.6 μm) in a gradient 2% B to 35% B in 90 min with a flow rate of 300 nL/min. Mobile phases were (A) 0.1% formic acid in water and (B) 0.1% formic acid in ACN. Sample ionizations were performed at a gas flow of 3.0 L/min, a temperature of 150 °C, and a capillary voltage of 1600 V. The quadrupole energy was fixed at 5.0 eV and the collision chamber energy at 7.0 eV, with an ion transfer time of 90 μs. Ions were analyzed in positive polarity mode in the range 150–2200 *m*/*z*, with an acquisition frequency of 1 Hz, and using the autoMS/MS system. The collected spectra were analyzed and calibrated using DataAnalysis software 6.1 (Bruker, Billerica, MA, USA) and then identified in ProteinScape 4 (Bruker) by the MASCOT server. Protein identification was conducted using the online SwissProt and NCBIprot databases, and their references and biological significance were identified using Reactome.org, String.org, and KEGG.

### 3.9. NO and PGE2 Production

RAW 264.7 cells were seeded (8 × 10^5^) in 6-well plates for 24 h, then exposed to LPS (1 μg/mL) or/and Selol (4 or 8 µg Se/mL) for 18 h. Untreated macrophages were taken as a control. The levels of NO and PGE2 were analyzed in the culture medium as previously described [69]. The absorbance at 540 nm was measured in a microplate reader BioTek, ELx 800 (BioTek Instruments, USA). The quantity of nitrite was determined from a sodium nitrite standard curve and expressed as μM. The PGE2 concentration was quantified using a competitive enzyme immunoassay kit PGE2 EIA Kit Cayman Chemical (Ann Arbor, MI, USA), according to the manufacturer’s instructions. The concentration of PGE2 was expressed as pg/mL.

### 3.10. Statistical Analysis

All data were representative of the experiments performed in triplicate and were expressed as the mean ± standard deviation (SD). The assessment of differences between groups was analyzed by Student’s *t*-test or two-way ANOVA, followed by the Tukey test. Differences were considered significant if the probability (*p*)-value was <0.05.

## 4. Conclusions

The results complete the knowledge of the molecular action of Se (IV), contained in Selol, both under physiological and inflammatory conditions. Based on our previous research [40], we hypothesize that Se (IV) in Selol has immunoreactive properties. The present results showed pro-oxidative properties of Selol in immune cells—RAW 264.7—and its ability to modulate cellular energy metabolism under physiological conditions. However, it reprograms cellular metabolism towards aerobic processes under inflammatory conditions. Selol showed weak anti-inflammatory effect via a decrease in the production of pro-inflammatory mediators and NF-κB activation. This may suggest that macrophages in the presence of Selol may change their action towards anti-inflammatory.

Se (IV) in Selol, which shows a dual effect—pro- and anti-oxidant properties—depending on the physiological/metabolic conditions of the cell, seems to be a promising element for the prevention and/or treatment of cancer/diseases with an inflammatory origin. In order to apply the conclusions drawn from the above experiments conducted on the in vitro macrophage cell model to changes in healthy and pathological tissues/organisms, extended studies are needed.

## Figures and Tables

**Figure 1 ijms-26-00559-f001:**
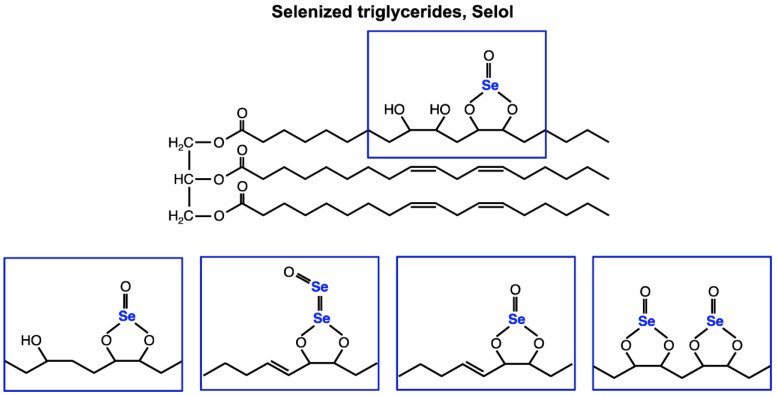
Chemical structure of Selol. In Selol, a complex mixture of at least 11 selenitetriglycerides compounds were characterized by high-performance liquid chromatography–electrospray ionization tandem mass spectrometry (HLPC-ESI MS^n^) [24].

**Figure 2 ijms-26-00559-f002:**
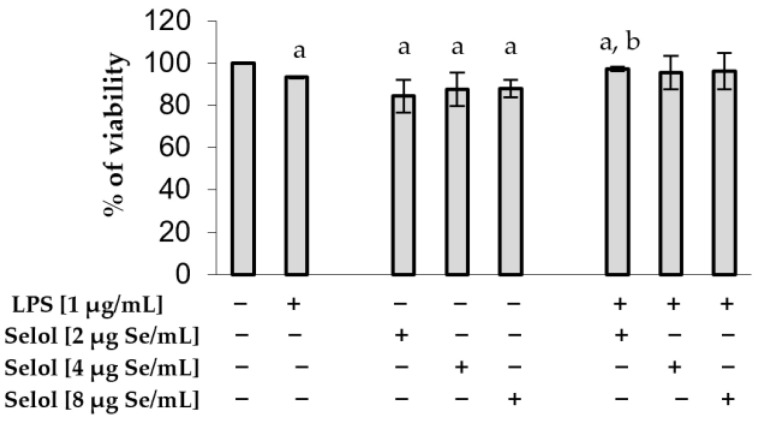
Cell viability of Selol-treated RAW 264.7 cells using the MTT test. Values are expressed as percentages of viable cells with respect to untreated (control) cells (+ means with compound, − means without compound). All data represent the means ± SD of three experiments, each of them performed in triplicate. ^a^
*p* < 0.05 versus control cells (Student’s *t*-test); ^b^
*p* < 0.05 versus LPS-treated cells (Student’s *t*-test).

**Figure 3 ijms-26-00559-f003:**
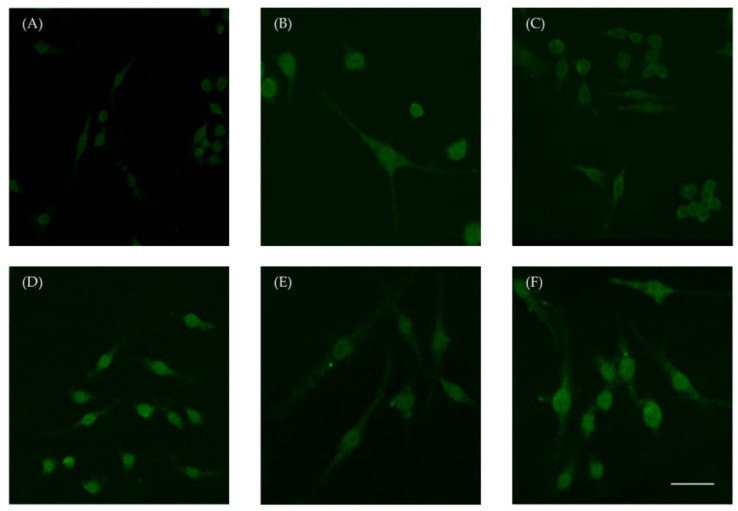
Representative confocal microscopy pictures demonstrating the subcellular localization of NF-κB/p65 in RAW 264.7 cells treated with Selol or/and with LPS for 1 h. (**A**) The untreated RAW 264.7 cells (control), (**B**) cells treated with Selol (4 μg Se/mL), (**C**) cells treated with Selol (8 μg Se/mL), (**D**) cells treated with LPS (1 μg/mL), (**E**) cells treated with Selol 4 μg Se/mL together with LPS (1 μg/mL), and (**F**) cells treated with Selol 8 μg Se/mL together with LPS (1 μg/mL). Results are representative of five independent experiments. Bar 20 nm.

**Figure 4 ijms-26-00559-f004:**
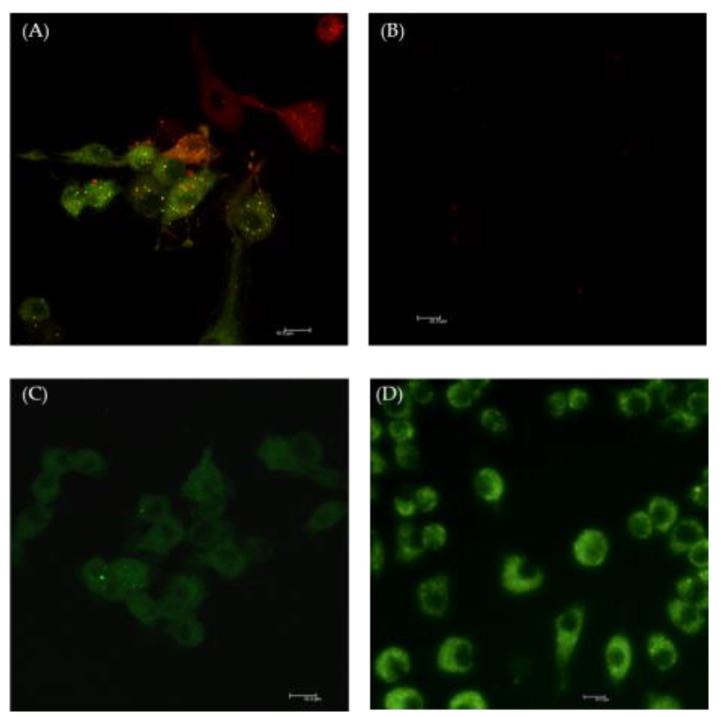
Representative confocal microscopy pictures of ROS production in RAW 264.7 cells treated with 4 or 8 μg Se/mL Selol for 1 h. ROS detection dye DHR 123 is shown as green fluorescence; dead cells are stained with propidine iodide (red fluorescence). (**A**) positive control, 1.5 mM H_2_O_2_, (**B**) control, untreated cells, (**C**) Selol 4 μg Se/mL, and (**D**) Selol 8 μg Se/mL.

**Figure 5 ijms-26-00559-f005:**
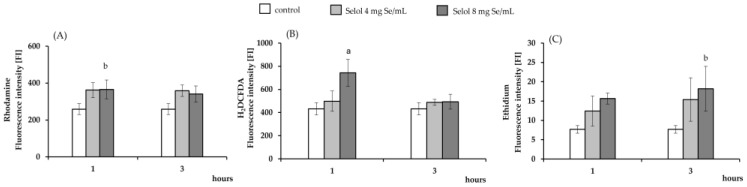
Level of ROS in RAW 264.7 cells stimulated with Selol. (**A**) Fluorescence intensity (FI) of the probe Rhodamine (1 μM), (**B**) fluorescence intensity of the probe DCF (5 µM), and (**C**) fluorescence intensity of the probe Ethidium (5 μM) in the presence of Selol (4 or 8 μg Se/mL) for 1 or 3 h. The results are expressed as mean ± SD of three experiments, each of them performed in triplicate. ^a^
*p* < 0.02 versus control cells (two-way ANOVA test); ^b^
*p* < 0.05 versus control cells (two-way ANOVA test).

**Figure 6 ijms-26-00559-f006:**
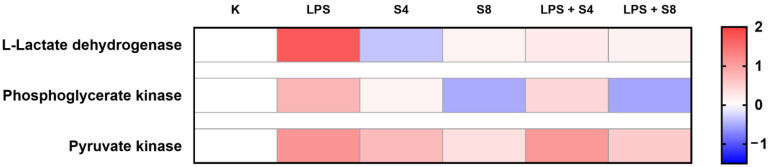
The normalized heat map of three interested proteins: L-lactate dehydrogenase A chain OS = Mus musculus OX = 10,090 GN = Ldha PE = 1 SV = 3; Phosphoglycerate kinase 1 OS = Mus musculus OX = 10,090 GN = Pgk1 PE = 1 SV = 4; and Pyruvate kinase PKM OS = Mus musculus OX = 10,090 GN = Pkm PE = 1 SV = 4. Red indicates a high expression level; blue indicates a low expression level. *p* < 0.05 versus control cells.

**Figure 7 ijms-26-00559-f007:**
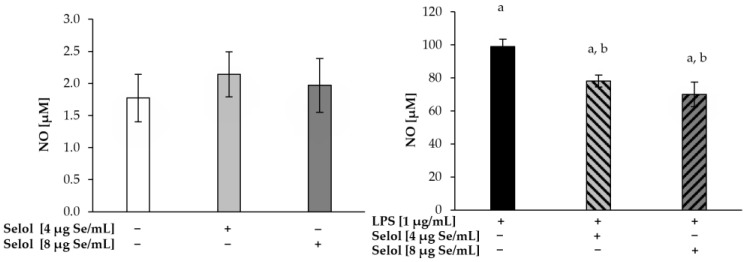
NO production in RAW 264.7 cells exposed to 4 or 8 μg Se/mL Selol or/and LPS for 18 h. (+ means with compound, − means without compound). Values are the mean ± SD of three experiments, each of them performed in triplicate. ^a^ *p* < 0.001 versus control cells (Student’s *t*-test); ^b^
*p* < 0.001 versus LPS-treated cells (Student’s *t*-test).

**Figure 8 ijms-26-00559-f008:**
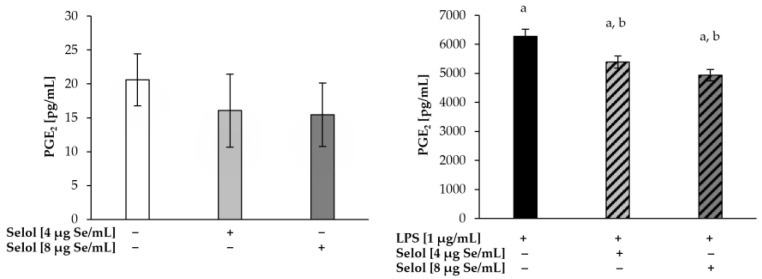
PGE_2_ production in RAW 264.7 cells exposed to 4 or 8 μg Se/mL Selol or/and LPS for 18 h. (+ means with compound, − means without compound). Values are the mean ± SD of three experiments, each of them performed in triplicate. ^a^ *p* < 0.001 versus control cells (Student’s *t*-test); ^b^
*p* < 0.001 versus LPS-treated cells (Student’s *t*-test).

**Table 1 ijms-26-00559-t001:** Effect of Selol and/or LPS on GSH_t_, GSSG, and thioredoxin level, as well as GSH/GSSG ratio in RAW 264.7 cells.

	GSH_t_ [nmoL/10^6^ Cells]	GSSG [nmoL/10^6^ Cells]	GSH/GSSG	Thioredoxin [Protein Intensity, %]
Control	10.48 ± 1.49	0.13 ± 0.0071	81.52 ± 2.44	100
S4	10.49 ± 0.70	0.13 ± 0.0063	75.72 ± 2.83	177.36
S8	10.75 ± 0.99	0.12 ± 0.0059	84.56 ± 4.99	214.55
LPS	2.06 ± 0.48 ^a^	0.18 ± 0.0084 ^b^	9.36 ± 0.25 ^a^	235.97
LPS + S4	4.73 ± 0.64 ^b, c#^	0.14 ± 0.0076 ^c#^	30.78 ± 1.47 ^a, b#^	82.17
LPS + S8	5.47 ± 0.52 ^b, c#^	0.14 ± 0.01 ^c#^	37.50 ± 1.21 ^a, a#^	154.39

^a^ *p* < 0.001 versus control cells; ^b^
*p* < 0.02 versus control cells; ^a#^
*p* < 0.001 versus LPS-treated cells; ^b#^
*p* < 0.02 versus LPS-treated cells; ^c#^
*p* < 0.05 versus LPS-treated cells (Student’s *t*-test).

## Data Availability

Data will be made available on request.

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
