# Peer review of "The Modulatory Effect of Selol (Se IV) on Pro-Inflammatory Pathways in RAW 264.7 Macrophages"

_ijms, 2025, doi:10.3390/ijms26020559_

Round 1
Reviewer 1 Report
Comments and Suggestions for Authors
INTRODUCTION:
The first part of this section is too long and general (lines 1-97), i suggest to intoduce better Selol (Se IV) and the background around this compound.
MATERIAL AND METHODS:
Lines 338-340 , please mention more information about how obtained and stored the sunflower oil.
then you mention The batches of Selol (1 mg Se/mL) used for this study were character- 341 ized by analytical chemistry for molecular constituents
discuss more about that.........what are the findings...
Line 368, why 1.5 mM H2O2? reference that, you can';t just mention it.
372 to 374, its not clear to me why you used all the probes, explain
Conclusion: the conclusion lookes like a repetition of the results, pelase improve accordingly and present the real conclusion of your study
RESULTS:
Figure 4, please provide better images
Author Response
Thank you very much for taking the time to review our manuscript. Please find the detailed responses below and the corresponding corrections highlighted in the re-submitted files
Introduction: "The first part of this section is too long and general (lines 1-97), i suggest to intoduce better Selol (Se IV) and the background around this compound"
Thank your very much for this comment. In accordance with the recommendations, the introduction section has been shortened and new publications have been added which describe the biological properties of selol - The new veriosion of concluisons and new text about selol are written in red colour.
MATERIALS AND METHODS:
"Lines 338-340 , please mention more information about how obtained and stored the sunflower oil".
then you mention The batches of Selol (1 mg Se/mL) used for this study were character- 341 ized by analytical chemistry for molecular constituents; discuss more about that.........what are the findings...
This fragment has been modified as recommended. We have added links to publications that describe in detail the technique used to produce selol and the form in which it is used in experiments. The new text is signed in red. We are not specialists in the technical side of selol preparation. We used fresh micellar selol solution in our experiments. We hope that the above explanation is sufficient.
Line 368, why 1.5 mM H2O2? reference that, you can';t just mention it
Thank you for this comment. Administration of H2O2 at 1.5 mM gave a clear increase in fluorescence; We have choosed this concentration based published dates; Kristiansen, K.A., Jensen, P.E., Møller, I.M. and Schulz, A. (2009), Monitoring reactive oxygen species formation and localisation in living cells by use of the fluorescent probe CM-H2DCFDA and confocal laser microscopy. Physiologia Plantarum, 136: 369-383. https://doi.org/10.1111/j.1399-3054.2009.01243.x; Xu, J., Hao, Z., Gou, X., Tian, W., Jin, Y., Cui, S., Guo, J., Sun, Y., Wang, Y. and Xu, Z. (2013), Imaging of reactive oxygen species burst from mitochondria using laser scanning confocal microscopy. Microsc. Res. Tech., 76: 612-617. https://doi.org/10.1002/jemt.22207; Both are included in the revised verision of our manuscript.
372 to 374, its not clear to me why you used all the probes, explain
All the three mentioned dyes are necessary to detect the presences of reactive oxygens species. Both DCFH and Rhodamine are commonly used for H202 dectection whereas HE is specialized with detecting intracellular O2•−
Conclusion: the conclusion lookes like a repetition of the results, pelase improve accordingly and present the real conclusion of your study
Thank you for this recomendation. The section conclusions has been revised. The new one is written in red.
Figure 4, please provide better images
Thank you for this suggestion, we replaced old images with new one,
Reviewer 2 Report
Comments and Suggestions for Authors
This is a first revision for manuscript "The modulatory effect of Selol (Se IV) on pro-inflammatory pathways in RAW 264.7 macrophages (ijms-3328763)". The study investigated the ability of Selol compound to exert an anti-inflammatory effect and affect the energy metabolism in murine RAW 264.7 cells exposed to pro-inflammatory LPS. It is interesting, but some points need reinforcement:
General revision of writing and typing
- line 14: study
- line 16: verify the use of abbreviation the first time they are included in the text (LPS). This amendment should be carefully revised throughout the entire manuscript.
- Punctuation in line 29 (The same in line 455)
- line 116: past tense
- line 302: revise typing
- lines 344-351: revise the use of symbols
- line 368: was "used" as a positive control...
Lines 102-106: Cite the cell line used.
Figure: the font size of some numbers could be bigger
Results and Discussion: The interpretation and description of results regarding the ambiguous-controversial properties of Selol (pro-oxidative / anti-inflammatory) may benefit from discussion of data from a recent research article about the immunomodulatory role of another bioactive compound (DOI: 10.1002/mnfr.202001034). Differential results between in vitro cell lines, such as the murine macrophage RAW 264.7, and ex vivo human intestinal biopsies (also challenged by pro-inflammatory LPS) may be relevant.
Conclusions: There is a considerable amount of results, but the feeling of a mixture of data, without a single clear direction. The authors may need to reinterpret all the information in the final discussion and conclusion section. They could indicate limitations of the study and future perspectives to further clarify their study and highlight its novelty with respect to what has already been published on Selol.
References: There is a high number of references for a research article (71). Some of them are not really necessary. Moreover, some of them are kind of old (more than 10 years) and/or published in non-reputable journals. Some should be replaced by new ones and/or be eliminated in case of not being really needed
Comments on the Quality of English LanguageIt may be improved
Author Response
Thank you very much for your comments, we corrected our manuscript according to your suggestions. Please find the detailed responses below and the corresponding corrections highlighted in the re-submitted files
- line 14: study - has been corrected
- line 16: verify the use of abbreviation the first time they are included in the text (LPS). This amendment should be carefully revised throughout the entire manuscript. - has been checked and corrected
- Punctuation in line 29 (The same in line 455) - hase been corrrected
-
lines 344-351: revise the use of symbols - corrected
line 368: was "used" as a positive control... - corrected
Lines 102-106: Cite the cell line used - corrected
The interpretation and description of results regarding the ambiguous-controversial properties of Selol (pro-oxidative / anti-inflammatory) may benefit from discussion of data from a recent research article about the immunomodulatory role of another bioactive compound (DOI: 10.1002/mnfr.202001034). Differential results between in vitro cell lines, such as the murine macrophage RAW 264.7, and ex vivo human intestinal biopsies (also challenged by pro-inflammatory LPS) may be relevant. - Thank you for this recomendation. We agree with this and the above research has been included in the revised Manuscript, in the section titled "Effect of Selol on NO/PGE2 produciton" the new text is written in red colour
Conlusions: There is a considerable amount of results, but the feeling of a mixture of data, without a single clear direction. The authors may need to reinterpret all the information in the final discussion and conclusion section. They could indicate limitations of the study and future perspectives to further clarify their study and highlight its novelty with respect to what has already been published on Selol -
Thank you very much for this comment, we revised the enitire section of conclusions.
References: There is a high number of references for a research article (71). Some of them are not really necessary. Moreover, some of them are kind of old (more than 10 years) and/or published in non-reputable journals. Some should be replaced by new ones and/or be eliminated in case of not being really needed
We revised the section of references, we replaced some old for the new ones, and removed the oldest ones
Figure: the font size of some numbers could be bigger
We have modifed figures included in the mansurcipt.
Round 2
Reviewer 2 Report
Comments and Suggestions for Authors
The authors have successfully addressed most of the comments raised by previous reviewers
Comments on the Quality of English Language-